# Variance Reduction with Sparse Gradients

**Melih Elibol, Michael I. Jordan**
University of California, Berkeley
{elibol,jordan}@cs.berkeley.edu

**Lihua Lei**
Stanford University
lihualei@stanford.edu

## Abstract

Variance reduction methods such as SVRG (Johnson & Zhang, 2013) and SpiderBoost (Wang et al., 2018) use a mixture of large and small batch gradients to reduce the variance of stochastic gradients. Compared to SGD (Robbins & Monro, 1951), these methods require at least double the number of operations per update to model parameters. To reduce the computational cost of these methods, we introduce a new sparsity operator: The random-top-$k$ operator. Our operator reduces computational complexity by estimating gradient sparsity exhibited in a variety of applications by combining the top-$k$ operator (Stich et al., 2018; Aji & Heafield, 2017) and the randomized coordinate descent operator. With this operator, large batch gradients offer an extra benefit beyond variance reduction: A reliable estimate of gradient sparsity. Theoretically, our algorithm is at least as good as the best algorithm (SpiderBoost), and further excels in performance whenever the random-top-$k$ operator captures gradient sparsity. Empirically, our algorithm consistently outperforms SpiderBoost using various models on various tasks including image classification, natural language processing, and sparse matrix factorization. We also provide empirical evidence to support the intuition behind our algorithm via a simple gradient entropy computation, which serves to quantify gradient sparsity at every iteration.

## 1 Introduction

Optimization tools for machine learning applications seek to minimize the finite sum objective

$$\min_{x \in \mathbb{R}^d} f(x) \triangleq \frac{1}{n} \sum_{i=1}^{n} f_i(x), \tag{1}$$

where $x$ is a vector of parameters, and $f_i : \mathbb{R}^d \to \mathbb{R}$ is the loss associated with sample $i$. Batch SGD serves as the prototype for modern stochastic gradient methods. It updates the iterate $x$ with $x - \eta \nabla f_{\mathcal{I}}(x)$, where $\eta$ is the learning rate and $\nabla f_{\mathcal{I}}(x)$ is the batch stochastic gradient, i.e.

$$\nabla f_{\mathcal{I}}(x) = \frac{1}{|\mathcal{I}|} \sum_{i \in \mathcal{I}} \nabla f_i(x).$$

The batch size $|\mathcal{I}|$ in batch SGD directly impacts the stochastic variance and gradient query complexity of each iteration of the update rule.

In recent years, variance reduction techniques have been proposed by carefully blending large and small batch gradients (e.g. Roux et al., 2012; Johnson & Zhang, 2013; Defazio et al., 2014; Xiao & Zhang, 2014; Allen-Zhu & Yuan, 2016; Allen-Zhu & Hazan, 2016; Reddi et al., 2016a;b; Allen-Zhu, 2017; Lei & Jordan, 2017; Lei et al., 2017; Allen-Zhu, 2018b; Fang et al., 2018; Zhou et al., 2018; Wang et al., 2018; Pham et al., 2019; Nguyen et al., 2019; Lei & Jordan, 2019). They are alternatives to batch SGD and are provably better than SGD in various settings. While these methods allow for greater learning rates than batch SGD and have appealing theoretical guarantees, they require a per-iteration query complexity which is more than double than that of batch SGD. Defazio (2019) questions the utility of variance reduction techniques in modern machine learning problems, empirically identifying query complexity as one issue. In this paper, we show that gradient sparsity (Aji & Heafield, 2017) can be used to significantly reduce the query complexity of variance reduction methods. Our work is motivated by the observation that gradients tend to be "sparse," having only

a small fraction of large coordinates. Specifically, if the indices of large gradient coordinates (measured in absolute value) are known before updating model parameters, we compute the derivative of only those coordinates while setting the remaining gradient coordinates to zero. In principle, if sparsity is exhibited, using large gradient coordinates will not effect performance and will significantly reduce the number of operations required to update model parameters. Nevertheless, this heuristic alone has three issues: (1) bias is introduced by setting other entries to zero; (2) the locations of large coordinates are typically unknown; (3) accessing a subset of coordinates may not be easily implemented for some problems like deep neural networks.

We provide solutions for all three issues. First, we introduce a new sparse gradient operator: The random-top-$k$ operator. The random-top-$k$ operator is a composition of the randomized coordinate descent operator and the top-$k$ operator. In prior work, the top-$k$ operator has been used to reduce the communication complexity of distributed optimization (Stich et al., 2018; Aji & Heafield, 2017) applications. The random-top-$k$ operator has two phases: Given a stochastic gradient and a pair of integers $(k_1, k_2)$ that sum to $k$, the operator retains $k_1$ coordinates which are most "promising" in terms of their "likelihood" to be large on average, then randomly selects $k_2$ of the remaining coordinates with appropriate rescaling. The first phase captures sparsity patterns while the second phase eliminates bias. Second, we make use of large batch gradients in variance reduction methods to estimate sparsity patterns. Inspired by the use of a memory vector in Aji & Heafield (2017), the algorithm maintains a memory vector initialized with the absolute value of the large batch gradient at the beginning of each outer loop and updated by taking an exponential moving average over subsequent stochastic gradients. Coordinates with large values in the memory vector are more "promising," and the random-top-$k$ operator will pick the top $k_1$ coordinate indices based on the memory vector. Since larger batch gradients have lower variance, the initial estimate is quite accurate. Finally, for software that supports dynamic computation graphs, we provide a cost-effective way (sparse back-propagation) to implement the random-top-$k$ operator.

In this work we apply the random-top-$k$ operator to SpiderBoost (Wang et al., 2018), a recent variance reduction method that achieves optimal query complexity, with a slight modification based on the "geometrization" technique introduced by Lei & Jordan (2019). Theoretically, we show that our algorithm is never worse than SpiderBoost and can strictly outperform it when the random-top-$k$ operator captures gradient sparsity. Empirically, we demonstrate the improvements in computation for various tasks including image classification, natural language processing, and sparse matrix factorization.

The rest of the paper is organized as follows. In Section 2, we define the random-top-$k$ operator, our optimization algorithm, and a description of sparse backpropagation. The theoretical analyses are presented in Section 3, followed by experimental results in Section 4. All technical proofs are relegated to Appendix A, and additional experimental details can be found in Appendix B.

## 2    STOCHASTIC VARIANCE REDUCTION WITH SPARSE GRADIENTS

Generally, variance reduction methods reduce the variance of stochastic gradients by taking a snapshot $\nabla f(y)$ of the gradient $\nabla f(x)$ every $m$ steps of optimization, and use the gradient information in this snapshot to reduce the variance of subsequent smaller batch gradients $\nabla f_{\mathcal{I}}(x)$ (Johnson & Zhang, 2013; Wang et al., 2018). Methods such as SCSG (Lei & Jordan, 2017) utilize a large batch gradient, which is typically some multiple in size of the small batch gradient $b$, which is much more practical and is what we do in this paper. To reduce the cost of computing additional gradients, we use sparsity by only computing a subset $k$ of the total gradients $d$, where $y \in \mathbb{R}^d$.

For $d, k, k_1, k_2 \in \mathbb{Z}_+$, let $k = k_1 + k_2$, where $1 \leq k \leq d$ for a parametric model of dimension $d$. In what follows, we define an operator which takes vectors $x, y$ and outputs $y'$, where $y'$ retains only $k$ of the entries in $y$, $k_1$ of which are selected according to the coordinates in $x$ which have the $k_1$ largest absolute values, and the remaining $k_2$ entries are randomly selected from $y$. The $k_1$ coordinate indices and $k_2$ coordinate indices are disjoint. Formally, the operator $\mathrm{rtop}_{k_1,k_2} : \mathbb{R}^d \to \mathbb{R}^d$ is defined for $x, y \in \mathbb{R}^d$ as

$$\left(\mathrm{rtop}_{k_1,k_2}(x,y)\right)_\ell = \begin{cases} y_\ell & \text{if } k_1 > 0 \text{ and } |x|_\ell \geq |x|_{(k_1)} \\ \frac{(d-k_1)}{k_2} y_\ell & \text{if } \ell \in S \\ 0 & \text{otherwise,} \end{cases}$$

where $|x|$ denotes a vector of absolute values, $|x|_{(1)} \geq |x|_{(2)} \geq \ldots \geq |x|_{(d)}$ denotes the order statistics of coordinates of $x$ in absolute values, and $S$ denotes a random subset with size $k_2$ that is uniformly drawn from the set $\{\ell : |x|_\ell < |x|_{(k_1)}\}$. For instance, if $x = (11, 12, 13, -14, -15), y = (-25, -24, 13, 12, 11)$ and $k_1 = k_2 = 1$, then $S$ is a singleton uniformly drawn from $\{1, 2, 3, 4\}$. Suppose $S = \{2\}$, then $\mathrm{rtop}_{1,1}(x, y) = (0, 4y_2, 0, 0, y_5) = (0, -96, 0, 0, 11)$. If $k_1 + k_2 = d$, $\mathrm{rtop}_{k_1,k_2}(x, y) = y$. On the other hand, if $k_1 = 0$, $\mathrm{rtop}_{0,k_2}(x, y)$ does not depend on $x$ and returns a rescaled random subset of $y$. This is the operator used in coordinate descent methods. Finally, $\mathrm{rtop}_{k_1,k_2}(x, y)$ is linear in $y$. The following Lemma shows that $\mathrm{rtop}_{k_1,k_2}(x, y)$ is an unbiased estimator of $y$, which is a crucial property in our later analysis.

**Lemma 1.** *Given any $x, y \in \mathbb{R}^d$,*

$$\mathbb{E}\left(\mathrm{rtop}_{k_1,k_2}(x, y)\right) = y, \quad \mathrm{Var}\left(\mathrm{rtop}_{k_1,k_2}(x, y)\right) = \frac{d - k_1 - k_2}{k_2} \|\mathrm{top}_{-k_1}(x, y)\|^2,$$

*where $\mathbb{E}$ is taken over the random subset $S$ involved in the $\mathrm{rtop}_{k_1,k_2}$ operator and*

$$(\mathrm{top}_{-k_1}(x, y))_\ell = \begin{cases} y_\ell & \text{if } k_1 > 0 \text{ and } |x|_\ell < |x|_{(k_1)} \\ 0 & \text{otherwise.} \end{cases}$$

Our algorithm is detailed as below.

---

**Algorithm 1:** SpiderBoost with Sparse Gradients.

**Input:** Learning rate $\eta$, inner loop size $m$, outer loop size $T$, large batch size $B$, small batch size $b$, initial iterate $x_0$, memory decay factor $\alpha$, sparsity parameters $k_1, k_2$.

1   $\mathcal{I}_0 \sim \mathrm{Unif}(\{1, \ldots, n\})$ with $|\mathcal{I}_0| = B$
2   $M_0 := |\nabla f_{\mathcal{I}_0}(x_0)|$
3   **for** $j = 1, \ldots, T$ **do**
4      $x_0^{(j)} := x_{j-1}, \; M_0^{(j)} := M_{j-1}$
5      $\mathcal{I}_j \sim \mathrm{Unif}(\{1, \ldots, n\})$ with $|\mathcal{I}_j| = B$
6      $\nu_0^{(j)} := \nabla f_{\mathcal{I}_j}(x_0^{(j)})$
7      $N_j := m$ (for implementation) or $N_j \sim$ geometric distribution with mean $m$ (for theory)
8      **for** $t = 0, \ldots, N_j - 1$ **do**
9          $x_{t+1}^{(j)} := x_t^{(j)} - \eta \nu_t^{(j)}$
10         $\mathcal{I}_t^{(j)} \sim \mathrm{Unif}([n])$ with $|\mathcal{I}_t^{(j)}| = b$
11        $\nu_{t+1}^{(j)} := \nu_t^{(j)} + \mathrm{rtop}_{k_1,k_2}\left(M_t^{(j)}, \nabla f_{\mathcal{I}_t^{(j)}}(x_{t+1}^{(j)}) - \nabla f_{\mathcal{I}_t^{(j)}}(x_t^{(j)})\right)$
12        $M_{t+1}^{(j)} := \alpha |\nu_{t+1}^{(j)}| + (1 - \alpha) M_t^{(j)}$
13      $x_j := x_{N_j}^{(j)}, \; M_j := M_{N_j}^{(j)}$

**Output:** $x_{\mathrm{out}} = x_T$ (for implementation) or $x_{\mathrm{out}} = x_{T'}$ where $T' \sim \mathrm{Unif}([T])$ (for theory)

---

The algorithm includes an outer-loop and an inner-loop. In the theoretical analysis, we generate $N_j$ as Geometric random variables. This trick is called "geometrization", proposed by Lei & Jordan (2017) and dubbed by Lei & Jordan (2019). It greatly simplifies analysis (e.g. Lei et al., 2017; Allen-Zhu, 2018a). In practice, as observed by Lei et al. (2017), setting $N_j$ to $m$ does not impact performance in any significant way. We only use "geometrization" in our theoretical analysis for clarity. Similarly, for our theoretical analysis, the output of our algorithm is selected uniformly at random from the set of outer loop iterations. Like the use of average iterates in convex optimization, this is a common technique for nonconvex optimization proposed by Nemirovski et al. (2009). In practice, we simply use the last iterate.

Similar to Aji & Heafield (2017), we maintain a memory vector $M_t^{(j)}$ at each iteration of our algorithm. The memory vector is initialized to the large batch gradient computed before every pass through the inner loop, which provides a relatively accurate gradient sparsity estimate of $x_0^{(j)}$. The exponential moving average gradually incorporates information from subsequent small batch gradients to account for changes to gradient sparsity. We then use $M_t^{(j)}$ as an approximation to the variance of each gradient coordinate in our $\mathrm{rtop}_{k_1,k_2}$ operator. With $M_t^{(j)}$ as input, the $\mathrm{rtop}_{k_1,k_2}$

operator targets $k_1$ high variance gradient coordinates in addition to $k_2$ randomly selected coordinates.

The cost of invoking $\text{rtop}_{k_1,k_2}$ is dominated by the algorithm for selecting the top $k$ coordinates, which has linear worst case complexity when using the `introselect` algorithm (Musser, 1997).

## 2.1 SPARSE BACK-PROPAGATION

A weakness of our method is the technical difficulty of implementing a sparse backpropagation algorithm in modern machine learning libraries, such as Tensorflow (Abadi et al., 2015) and Pytorch (Paszke et al., 2017). Models implemented in these libraries generally assume dense structured parameters. The optimal implementation of our algorithm makes use of a sparse forward pass and assumes a sparse computation graph upon which backpropagation is executed. Libraries that support dynamic computation graphs, such as Pytorch, will construct the sparse computation graph in the forward pass, which makes the required sparse backpropagation trivial. We therefore expect our algorithm to perform quite well on libraries which support dynamic computation graphs.

Consider the forward pass of a deep neural network, where $\phi$ is a deep composition of parametric functions,

$$\phi(x;\theta) = \phi_L(\phi_{L-1}(\ldots \phi_0(x;\theta_0)\ldots;\theta_{L-1});\theta_L). \tag{2}$$

The unconstrained problem of minimizing over the $\theta_\ell$ can be rewritten as a constrained optimization problem as follows:

$$\begin{aligned}
\min_\theta \quad & \frac{1}{n}\sum_{i=1}^n \text{loss}(z_i^{(L+1)}, y_i) \\
\text{s.t.} \quad & z_i^{(L+1)} = \phi_L(z_i^{(L)};\theta_L) \\
& \vdots \\
& z_i^{(\ell+1)} = \phi_\ell(z_i^{(\ell)};\theta_\ell) \\
& \vdots \\
& z_i^{(1)} = \phi_0(x_i;\theta_0).
\end{aligned} \tag{3}$$

In this form, $z_i^{L+1}$ is the model estimate for data point $i$. Consider $\phi_\ell(x;\theta_\ell) = \sigma(x^T\theta_\ell)$ for $1 \leq \ell < L$, $\phi_L$ be the output layer, and $\sigma$ be some subdifferentiable activation function. If we apply the $\text{rtop}_{k_1,k_2}$ operator per-layer in the forward-pass, with appropriate scaling of $k_1$ and $k_2$ to account for depth, we see that the number of multiplications in the forward pass is reduced to $k_1 + k_2$: $\sigma(\text{rtop}_{k_1,k_2}(v,x)^T \text{rtop}_{k_1,k_2}(v,\theta_\ell))$. A sparse forward-pass yields a computation graph for a $(k_1 + k_2)$-parameter model, and back-propagation will compute the gradient of the objective with respect to model parameters in linear time (Chauvin & Rumelhart, 1995).

## 3 THEORETICAL COMPLEXITY ANALYSIS

### 3.1 NOTATION AND ASSUMPTIONS

Denote by $\|\cdot\|$ the Euclidean norm and by $a \wedge b$ the minimum of $a$ and $b$. For a random vector $Y \in \mathbb{R}^d$,

$$\text{Var}(Y) = \sum_{i=1}^d \text{Var}(Y_i).$$

We say a random variable $N$ has a geometric distribution, $N \sim \text{Geom}(m)$, if $N$ is supported on the non-negative integers with

$$\mathbb{P}(N = k) = \gamma^k(1 - \gamma), \quad \forall k = 0, 1, \ldots,$$

for some $\gamma$ such that $\mathbb{E}N = m$. Here we allow $N$ to be zero to facilitate the analysis.

Assumption **A**1 on the smoothness of individual functions will be made throughout the paper.

**A**1  $f_i$ is differentiable with
$$\|\nabla f_i(x) - \nabla f_i(y)\| \le L\|x - y\|,$$
for some $L < \infty$ and for all $i \in \{1, \dots, n\}$.

As a direct consequence of assumption **A**1, it holds for any $x, y \in \mathbb{R}^d$ that

$$-\frac{L}{2}\|x - y\|^2 \le f_i(x) - f_i(y) - \langle \nabla f_i(y), x - y \rangle \le \frac{L}{2}\|x - y\|^2. \tag{4}$$

To formulate our complexity bounds, we define

$$f^* = \inf_x f(x), \quad \Delta_f = f(x_0) - f^*.$$

Further we define $\sigma^2$ as an upper bound on the expected norm of the stochastic gradients:

$$\sigma^2 = \sup_x \frac{1}{n} \sum_{i=1}^n \|\nabla f_i(x)\|^2. \tag{5}$$

By Cauchy-Schwarz inequality, it is easy to see that $\sigma^2$ is also a uniform bound of $\|\nabla f(x)\|^2$.

Finally, we assume that sampling an index $i$ and accessing the pair $\nabla f_i(x)$ incur a unit of cost and accessing the truncated version $\mathrm{rtop}_{k_1, k_2}(m, \nabla f_i(x))$ incur $(k_1 + k_2)/d$ units of cost. Note that calculating $\mathrm{rtop}_{k_1, k_2}(m, \nabla f_{\mathcal{I}}(x))$ incurs $|\mathcal{I}|(k_1 + k_2)/d$ units of computational cost. Given our framework, the theoretical complexity of the algorithm is

$$C_{\mathrm{comp}}(\epsilon) \triangleq \sum_{j=1}^T \left( B + 2bN_j \frac{k_1 + k_2}{d} \right). \tag{6}$$

### 3.2  Worst-case guarantee

**Theorem 1.** *Under the following setting of parameters*

$$\eta L = \sqrt{\frac{k_2}{6dm}}, \quad B = \left\lceil \frac{2\sigma^2}{\epsilon^2} \wedge n \right\rceil$$

*For any $T \ge T(\epsilon) \triangleq 4\Delta_f / \eta m \epsilon^2$,*
$$\mathbb{E}\|\nabla f(x_{\mathrm{out}})\| \le \epsilon.$$

*If we further set*
$$m = \frac{Bd}{b(k_1 + k_2)},$$
*the complexity to achieve the above condition is*

$$\mathbb{E}C_{\mathrm{comp}}(\epsilon) = O\left( \left( \frac{\sigma}{\epsilon^3} \wedge \frac{\sqrt{n}}{\epsilon^2} \right) L\Delta_f \sqrt{\frac{b(k_1 + k_2)}{k_2}} \right).$$

Recall that the complexity of SpiderBoost (Wang et al., 2018) is

$$O\left( \left( \frac{\sigma}{\epsilon^3} \wedge \frac{\sqrt{n}}{\epsilon^2} \right) L\Delta_f \right).$$

Thus as long as $b = O(1), k_1 = O(k_2)$, our algorithm has the same complexity as SpiderBoost under appropriate settings. The penalty term $O(\sqrt{b(k_1 + k_2)/k_2})$ is due to the information loss by sparsification.

### 3.3 DATA ADAPTIVE ANALYSIS

Let

$$g_t^{(j)} = \| \operatorname{top}_{-k_1}(M_t^{(j)}, \nabla f(x_{t+1}^{(j)}) - \nabla f(x_t^{(j)}))\|^2,$$

and

$$G_t^{(j)} = \frac{1}{n}\sum_{i=1}^{n} \| \operatorname{top}_{-k_1}(M_t^{(j)}, \nabla f_i(x_{t+1}^{(j)}) - \nabla f_i(x_t^{(j)}))\|^2.$$

By Cauchy-Schwarz inequality and the linearity of $\operatorname{top}_{-k_1}$, it is easy to see that $g_t^{(j)} \leq G_t^{(j)}$. If our algorithm succeeds in capturing sparsity, both $g_t^{(j)}$ and $G_t^{(j)}$ will be small. In this subsection we will analyze the complexity under this case. Further define $R_j$ as

$$R_j = \mathbb{E}_j g_{N_j}^{(j)} + \frac{\mathbb{E}_j G_{N_j}^{(j)}}{b}, \tag{7}$$

where $\mathbb{E}_j$ is taken over all randomness in $j$-th outer loop (line 4-13 of Algorithm 1).

**Theorem 2.** *Under the following setting of parameters*

$$\eta L = \sqrt{\frac{b \wedge m}{3m}}, \quad B = \left\lceil \frac{3\sigma^2}{\epsilon^2} \wedge n \right\rceil$$

*For any $T \geq T(\epsilon) \triangleq 6\Delta_f/\eta m\epsilon^2$,*

$$\mathbb{E}\|\nabla f(x_{\mathrm{out}})\|^2 \leq \frac{2\epsilon^2}{3} + \frac{(d - k_1 - k_2)m}{k_2}\mathbb{E}\bar{R}_T,$$

*where*

$$\bar{R}_T = \frac{1}{T}\sum_{j=1}^{T} R_j.$$

*If $\mathbb{E}\bar{R}_T \leq \epsilon^2 \frac{k_2}{3(d - k_1 - k_2)m}$, then*

$$\mathbb{E}\|\nabla f(x_{\mathrm{out}})\| \leq \epsilon.$$

*If we further set*

$$m = \frac{Bd}{b(k_1 + k_2)},$$

*the complexity to achieve the above condition is*

$$\mathbb{E}C_{\mathrm{comp}}(\epsilon) = O\left(\left(\frac{\sigma}{\epsilon^3} \wedge \frac{\sqrt{n}}{\epsilon^2}\right) L\Delta_f \sqrt{\frac{k_1 + k_2}{d}\frac{b}{b \wedge m}}\right).$$

In practice, $m$ is usually much larger than $b$. As a result, the complexity of our algorithm is $O(\sqrt{(k_1 + k_2)/d})$ smaller than that of SpiderBoost if our algorithm captures gradient sparsity. Although this type of data adaptive analyses is not as clean as the worst-case guarantee (Theorem 1), it reveals the potentially superior performance of our algorithm. Similar analyses have been done for various other algorithms, including AdaGrad (Duchi et al., 2011) and Adam (Kingma & Ba, 2014).

## 4 EXPERIMENTS

In this section, we present a variety of experiments to illustrate gradient sparsity and demonstrate the performance of Sparse SpiderBoost. By computing the entropy of the empirical distribution of the absolute value of stochastic gradient coordinates, we show that certain models exhibit gradient sparsity during optimization. To evaluate the performance of variance reduction with sparse gradients, we compute the loss over gradient queries per epoch of Sparse Spiderboost and SpiderBoost for a number of image classification problems. We also compare Sparse SpiderBoost, SpiderBoost, and SGD on a natural language processing task and sparse matrix factorization.

For all experiments, unless otherwise specified, we run SpiderBoost and Sparse SpiderBoost with a learning rate $\eta = 0.1$, large-batch size $B = 1000$, small-batch size $b = 100$, inner loop length of $m = 10$, memory decay factor of $\alpha = 0.5$, and $k_1$ and $k_2$ both set to $5\%$ of the total number of model parameters. We call the sum $k_1 + k_2 = k = 10\%$ the sparsity of the optimization algorithm.

## 4.1 GRADIENT SPARSITY AND IMAGE CLASSIFICATION

Our experiments in this section test a number of image classification tasks for gradient sparsity, and plot the learning curves of some of these tasks. We test a 2-layer fully connected neural network with hidden layers of width 100, a simple convolutional neural net which we describe in detail in Appendix B, and Resnet-18 (He et al., 2015). All models use ReLu activations. For datasets, we use CIFAR-10 (Krizhevsky et al.), SVHN (Netzer et al., 2011), and MNIST (LeCun & Cortes, 2010). None of our experiments include Resnet-18 on MNIST as MNIST is an easier dataset; it is included primarily to provide variety.

Our method relies partially on the assumption that the magnitude of the derivative of some model parameters are greater than others. To measure this, we compute the entropy of the empirical distribution of the absolute value of stochastic gradient coordinates. In Algorithm 1, the following term updates our estimate of the variance of each coordinate's derivative:

$$M_{t+1}^{(j)} := \alpha |\nu_{t+1}^{(j)}| + (1 - \alpha) M_t^{(j)}.$$

Consider the entropy of the following probability vector $p_t^{(j)} = M_t^{(j)} / \|M_t^{(j)}\|_1$. The entropy of $p$ provides us with a measure of how much structure there is in our gradients. To see this, consider the hypothetical scenario where $p_i = 1/d$. In this scenario we have no structure; the top $k_1$ component of our sparsity operator is providing no value and entropy is maximized. On the other hand, if a single entry $p_i = 1$ and all other entries $p_j = 0$, then the top $k_1$ component of our sparsity operator is effectively identifying the only relevant model parameter.

To measure the potential of our sparsity operator, we compute the entropy of $p$ while running SpiderBoost on a variety of datasets and model architectures. The results of running this experiment are summarized in the following table.

Table 1: Entropy of Memory Vectors

|  | FC NN | | | Conv NN | | | Resnet-18 | | |
| --- | --- | --- | --- | --- | --- | --- | --- | --- | --- |
|  | Max | Before | After | Max | Before | After | Max | Before | After |
| CIFAR-10 | 18.234 | 16.41 | 8.09 | 15.920 | 13.38 | 2.66 | 23.414 | 22.59 | 21.70 |
| SVHN | 18.234 | 15.36 | 8.05 | 15.920 | 13.00 | 2.97 | 23.414 | 22.62 | 21.31 |
| MNIST | 18.234 | 14.29 | 9.77 | 15.920 | 14.21 | 2.77 | - | - | - |

Table 1 provides the maximum entropy as well as the entropy of the memory vector before and after training for 150 epochs, for each dataset and each model. For each model, the entropy at the beginning of training is almost maximal. This is due to random initialization of model parameters. After 150 epochs, the entropy of $M_t$ for the convolutional model drops to approximately 3, which suggests a substantial amount of gradient structure. Note that for the datasets that we tested, the gradient structure depends primarily on the model and not the dataset. In particular, for Resnet-18, the entropy appears to vary minimally after 150 epochs.

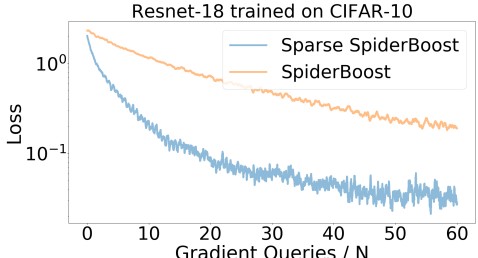 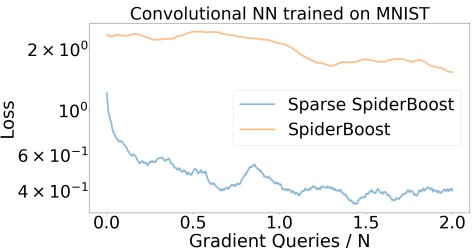

Figure 1: SpiderBoost with $10\%$ sparsity ($k = 0.1d$) compared to SpiderBoost without sparsity. Left figure compares the two algorithms using Resnet-18 on Cifar-10. Right figure compares the two algorithms using a convolutional neural network trained on MNIST. The x-axis measures gradient queries over $N$, where $N$ is the size of the respective datasets. Plots are in log-scale.

Figure 1 compares SpiderBoost alone to SpiderBoost with $10\%$ sparsity ($10\%$ of parameter derivatives). All experiments in this section are run for 50 epochs. In our comparison to SpiderBoost, we measure the number of gradient queries over the size of the dataset $N$. A single gradient query is taken to be the cost of computing a gradient for a single data point. If $i$ is the index of a single sample, then $\nabla f_i(x)$ is a single gradient query. Using the batch gradient to update model parameters for a dataset of size $B$ has a gradient query cost of $B$. For a model with $d$ parameters, using a single sample to update $k$ model parameters has a gradient query cost of $k/d$, etc.

Our results of fitting the convolutional neural network to MNIST show that sparsity provides a significant advantage compared to using SpiderBoost alone. We only show 2 epochs of this experiment since the MNIST dataset is fairly simple and convergence is rapidly achieved. The results of training Resnet-18 on CIFAR-10 suggests that our sparsity algorithm works well on large neural networks, and non-trivial datasets. We believe Resnet-18 on CIFAR-10 does not do as well due to the gradient density we observe for Resnet-18 in general. Sparsity here not only has the additional benefit of reducing gradient query complexity, but also provides a dampening effect on variance due to the additional covariates in SpiderBoost's update to model parameters. Results for the rest of these experiments can be found in Appendix B.

## 4.2 NATURAL LANGUAGE PROCESSING

We evaluate Sparse SpiderBoost's performance on an LSTM-based (Hochreiter & Schmidhuber, 1997) generative language model. We compare Sparse SpiderBoost, SpiderBoost, and SGD. We train our LSTM model on the Penn Treebank (Marcus et al., 1994) corpus. The natural language processing model consists of a word embedding of dimension 128 of 1000 tokens, which is jointly learned with the task. The LSTM has a hidden and cell state dimension of 1024. All three optimization algorithms operate on this model. The variance reduction training algorithm for this type of model can be found in Appendix B. We run SpiderBoost and Sparse SpiderBoost with a learning rate $\eta = 0.2$, large-batch size $B = 40$, small-batch size $b = 20$, inner loop length of $m = 2$. We run SGD with learning rate 0.2 and batch size is 20. Figure 2 shows SpiderBoost is slightly worse than SGD, and sparsity provides a noticeable improvement over SGD.

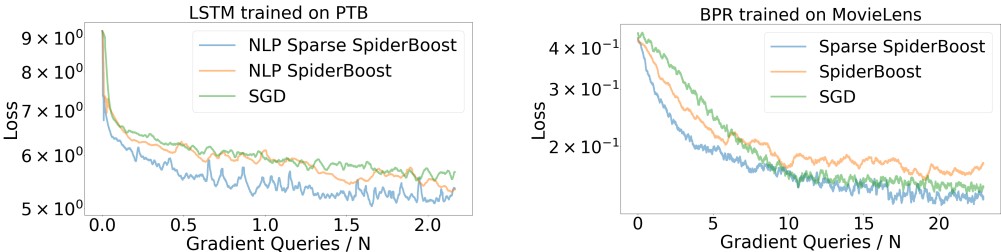

Figure 2: (a): SGD learning rate is 0.2 and batch size is 20. (b): SGD batch size is 103 and learning rate schedule is 0.1 for epochs $0 - 10$, 0.01 for epochs $10 - 20$, and 0.001 for epochs $20 - 40$. The x-axis measures gradient queries over $N$, where $N$ is the size of the respective datasets. Plots are in log-scale.

## 4.3 SPARSE MATRIX FACTORIZATION

For our experiments with sparse matrix factorization, we perform Bayesian Personalized Ranking (Rendle et al., 2009) on the MovieLens database (Harper & Konstan, 2015) with a latent dimension of 20. To satisfy $m = B/b$, we run SpiderBoost and Sparse SpiderBoost with a large-batch size $B = 1030$, small-batch size $b = 103$, inner loop length of $m = 10$. For this experiment, we run SpiderBoost with the following learning rate schedule:

$$\eta(a, b, t) = b + (a - b)\frac{m - t}{m},$$

where $a = 1.0$ and $b = 0.1$. The schedule interpolates from $a$ to $b$ as the algorithm progresses through the inner loop. For instance, within the inner loop, at iteration 0 the learning rate is 1.0, and at iteration $m$ the learning rate is 0.1. We believe this is a natural way to utilize the low variance

at the beginning of the inner loop, and is a fair comparison to an exponential decay learning rate schedule for SGD. Details of the SGD baselines are provided in Figure 2. We see SpiderBoost is slightly worse than SGD, and sparsity provides a slight improvement over SGD, especially in the first few epochs.

## 5 CONCLUSION

In this paper, we show how sparse gradients with memory can be used to improve the gradient query complexity of SVRG-type variance reduction algorithms. While we provide a concrete sparse variance reduction algorithm for SpiderBoost, the techniques developed in this paper can be adapted to other variance reduction algorithms.

We show that our algorithm provides a way to explicitly control the gradient query complexity of variance reduction methods, a problem which has thus far not been addressed. Assuming our algorithm captures the sparsity structure of the optimization problem, we also prove that the complexity of our algorithm is an improvement over SpiderBoost. The results of our comparison to Spider-Boost validates this assumption, and entropy measures provided in Table 1 empirically support our hypothesis that gradient sparsity exists.

Table 1 also supports the results in Aji & Heafield (2017), which shows that the top-$k$ operator generally outperforms the random-$k$ operator. Our random-top-$k$ operator takes advantage of the superior performance of the top-$k$ operator while eliminating bias via a secondary random-$k$ operator. Not every problem we tested exhibited sparsity structure. While this is true, our analysis proves that our algorithm performs no worse than SpiderBoost in these settings. Even when there is no structure, our algorithm reduces to a random sampling of $k_1 + k_2$ coordinates, which is essentially a randomized coordinate descent analogue of SpiderBoost. Empirically, we see that Sparse Spider-Boost outperforms SpiderBoost when no sparsity structure is present. We believe this is due to the variance introduced by additional covariates in the SpiderBoost update, which is mitigated in Sparse SpiderBoost by our random-top-$k$ operator.

The results of our experiments on natural language processing and matrix factorization demonstrate that, with additional effort, variance reduction methods are competitive with SGD. While we view this as progress toward improving the practical viability of variance reduction algorithms, we believe further improvements can be made, such as better utilization of reduced variance during training, and better control over increased variance in very high dimensional models such as dense net (Defazio, 2019). We recognize these issues and hope to make progress on them in future work.

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

# A    TECHNICAL PROOFS

## A.1    PREPARATORY RESULTS

**Lemma 2** (Lemma 3.1 of Lei & Jordan (2019)). *Let $N \sim \mathrm{Geom}(m)$. Then for any sequence $D_0, D_1, \dots$ with $\mathbb{E}|D_N| < \infty$,*

$$\mathbb{E}(D_N - D_{N+1}) = \frac{1}{m}(D_0 - \mathbb{E}D_N).$$

**Remark 1.** *The requirement $\mathbb{E}|D_N| < \infty$ is essential. A useful sufficient condition if $|D_t| = O(\mathrm{Poly}(t))$ because a geometric random variable has finite moments of any order.*

**Lemma 3** (Lemma B.2 of Lei & Jordan (2019)). *Let $z_1, \dots, z_M \in \mathbb{R}^d$ be an arbitrary population and $\mathcal{J}$ be a uniform random subset of $[M]$ with size $m$. Then*

$$\mathrm{Var}\left(\frac{1}{m}\sum_{j \in \mathcal{J}} z_j\right) \le \frac{I(m < M)}{m} \cdot \frac{1}{M}\sum_{j=1}^{M}\|z_j\|_2^2.$$

***Proof of Lemma 1.*** WLOG, assume that $|x_1| \ge |x_2| \ge \dots \ge |x_d|$. Let $S$ be a random subset of $\{k_1 + 1, \dots, d\}$ with size $k_2$. Then

$$\left(\mathrm{rtop}_{k_1,k_2}(x,y)\right)_\ell = y_\ell\left(I(\ell \le k_1) + \frac{d - k_1}{k_2}I(\ell \in S)\right).$$

As a result,

$$\mathbb{E}\left[\left(\mathrm{rtop}_{k_1,k_2}(x,y)\right)_\ell\right] = y_\ell\left(I(\ell \le k_1) + \frac{d - k_1}{k_2}I(\ell > k_1)P(\ell \in S)\right) = y_\ell,$$

and

$$\mathrm{Var}\left[\left(\mathrm{rtop}_{k_1,k_2}(x,y)\right)_\ell\right] = \left(\frac{d - k_1}{k_2}\right)^2 y_\ell^2 I(\ell > k_1)P(\ell \in S)(1 - P(\ell \in S))$$

$$= \frac{d - k_1 - k_2}{k_2}y_\ell^2 I(\ell > k_1).$$

Therefore,

$$\mathrm{Var}\left(\mathrm{rtop}_{k_1,k_2}(x,y)\right) = \frac{d - k_1 - k_2}{k_2}\sum_{\ell > k_1} y_\ell^2 = \frac{d - k_1 - k_2}{k_2}\|\mathrm{top}_{-k_1}(x,y)\|^2.$$

$\square$

## A.2    ANALYSIS OF A SINGLE INNER LOOP

**Lemma 4.** *For any $j, t$,*

$$\mathbb{E}_{j,t}(\nu_{t+1}^{(j)} - \nu_t^{(j)}) = \nabla f(x_{t+1}^{(j)}) - \nabla f(x_t^{(j)})$$

*and*

$$\mathrm{Var}_{j,t}(\nu_{t+1}^{(j)} - \nu_t^{(j)}) \le \frac{\eta^2 L^2}{b}\|\nu_t^{(j)}\|^2 + \frac{d - k_1 - k_2}{k_2}\left(g_t^{(j)} + \frac{G_t^{(j)}}{b}\right),$$

*where $\mathbb{E}_{j,t}$ and $\mathrm{Var}_{j,t}$ are taken over the randomness of $\mathcal{I}_t^{(j)}$ and the random subset $S$ involved in the $\mathrm{rtop}_{k_1,k_2}$ operator.*

*Proof.* By definition,

$$\nu_{t+1}^{(j)} - \nu_t^{(j)} = \mathrm{rtop}_{k_1,k_2}\left(M_t^{(j)}, \nabla f_{\mathcal{I}_t^{(j)}}(x_{t+1}^{(j)}) - \nabla f_{\mathcal{I}_t^{(j)}}(x_t^{(j)})\right).$$

Let $S$ be the random subset involved in $\mathrm{rtop}_{k_1,k_2}$. Then $S$ is independent of $(\mathcal{I}_t^{(j)}, M_t^{(j)}, x_{t+1}^{(j)}, x_t^{(j)})$. By Lemma 1,

$$\mathbb{E}_S\left(\nu_{t+1}^{(j)} - \nu_t^{(j)}\right) = \nabla f_{\mathcal{I}_t^{(j)}}(x_{t+1}^{(j)}) - \nabla f_{\mathcal{I}_t^{(j)}}(x_t^{(j)})$$

and

$$\mathrm{Var}_S\left(\nu_{t+1}^{(j)} - \nu_t^{(j)}\right) = \frac{d - k_1 - k_2}{k_2}\left\|\mathrm{top}_{-k_1}\left(M_t^{(j)}, \nabla f_{\mathcal{I}_t^{(j)}}(x_{t+1}^{(j)}) - \nabla f_{\mathcal{I}_t^{(j)}}(x_t^{(j)})\right)\right\|^2.$$

Since $\mathcal{I}_t^{(j)}$ is independent of $(M_t^{(j)}, x_{t+1}^{(j)}, x_t^{(j)})$, the tower property of conditional expectation and variance implies that

$$\mathbb{E}_{j,t}\left(\nu_{t+1}^{(j)} - \nu_t^{(j)}\right) = \mathbb{E}_{\mathcal{I}_t^{(j)}}\left(\nabla f_{\mathcal{I}_t^{(j)}}(x_{t+1}^{(j)}) - \nabla f_{\mathcal{I}_t^{(j)}}(x_t^{(j)})\right) = \nabla f(x_{t+1}^{(j)}) - \nabla f(x_t^{(j)}),$$

and

$$\mathrm{Var}_{j,t}\left(\nu_{t+1}^{(j)} - \nu_t^{(j)}\right) = \mathbb{E}_{\mathcal{I}_t^{(j)}}\left(\mathrm{Var}_S\left(\nu_{t+1}^{(j)} - \nu_t^{(j)}\right)\right) + \mathrm{Var}_{\mathcal{I}_t^{(j)}}\left(\mathbb{E}_S\left(\nu_{t+1}^{(j)} - \nu_t^{(j)}\right)\right). \quad (8)$$

To bound the first term, we note that $\mathrm{top}_{-k_1}$ is linear in $y$ and thus

$$\mathbb{E}_{\mathcal{I}_t^{(j)}}\left\|\mathrm{top}_{-k_1}\left(M_t^{(j)}, \nabla f_{\mathcal{I}_t^{(j)}}(x_{t+1}^{(j)}) - \nabla f_{\mathcal{I}_t^{(j)}}(x_t^{(j)})\right)\right\|^2$$

$$= \left\|\mathbb{E}_{\mathcal{I}_t^{(j)}}\mathrm{top}_{-k_1}\left(M_t^{(j)}, \nabla f_{\mathcal{I}_t^{(j)}}(x_{t+1}^{(j)}) - \nabla f_{\mathcal{I}_t^{(j)}}(x_t^{(j)})\right)\right\|^2$$

$$+ \mathrm{Var}_{\mathcal{I}_t^{(j)}}\left[\mathrm{top}_{-k_1}\left(M_t^{(j)}, \nabla f_{\mathcal{I}_t^{(j)}}(x_{t+1}^{(j)}) - \nabla f_{\mathcal{I}_t^{(j)}}(x_t^{(j)})\right)\right]$$

$$= g_t^{(j)} + \mathrm{Var}_{\mathcal{I}_t^{(j)}}\left[\frac{1}{b}\sum_{i\in\mathcal{I}_t^{(j)}}\mathrm{top}_{-k_1}(M_t^{(j)}, \nabla f_i(x_{t+1}^{(j)}) - \nabla f_i(x_t^{(j)}))\right]$$

$$\leq g_t^{(j)} + \frac{G_t^{(j)}}{b}, \quad (9)$$

where the last inequality uses Lemma 3. To bound the second term of (8), by Lemma 3,

$$\mathrm{Var}_{\mathcal{I}_t^{(j)}}\left(\mathbb{E}_S\left(\nu_{t+1}^{(j)} - \nu_t^{(j)}\right)\right) = \mathrm{Var}_{\mathcal{I}_t^{(j)}}\left(\nabla f_{\mathcal{I}_t^{(j)}}(x_{t+1}^{(j)}) - \nabla f_{\mathcal{I}_t^{(j)}}(x_t^{(j)})\right)$$

$$\leq \frac{1}{b}\frac{1}{n}\sum_{i=1}^n\|\nabla f_i(x_{t+1}^{(j)}) - \nabla f_i(x_t^{(j)})\|^2 \overset{(i)}{\leq} \frac{L^2}{b}\|x_{t+1}^{(j)} - x_t^{(j)}\|^2 \overset{(ii)}{=} \frac{\eta^2 L^2}{b}\|\nu_t^{(j)}\|^2,$$

where (i) uses assumption **A**1 and (ii) uses the definition that $x_{t+1}^{(j)} = x_t^{(j)} - \eta\nu_t^{(j)}$. $\qquad\square$

**Lemma 5.** *For any $j, t$,*

$$\mathbb{E}_{j,t}\|\nu_{t+1}^{(j)} - \nabla f(x_{t+1}^{(j)})\|^2 \leq \|\nu_t^{(j)} - \nabla f(x_t^{(j)})\|^2 + \frac{\eta^2 L^2}{b}\|\nu_t^{(j)}\|^2 + \frac{d - k_1 - k_2}{k_2}\left(g_t^{(j)} + \frac{G_t^{(j)}}{b}\right),$$

*where $\mathbb{E}_{j,t}$ and $\mathrm{Var}_{j,t}$ are taken over the randomness of $\mathcal{I}_t^{(j)}$ and the random subset $S$ involved in the $\mathrm{rtop}_{k_1,k_2}$ operator.*

*Proof.* By Lemma 4, we have

$$\nu_{t+1}^{(j)} - \nabla f(x_{t+1}^{(j)}) = \nu_t^{(j)} - \nabla f(x_t^{(j)}) + \left(\nu_{t+1}^{(j)} - \nu_t^{(j)} - \mathbb{E}_{j,t}(\nu_{t+1}^{(j)} - \nu_t^{(j)})\right).$$

Since $\mathcal{I}_t^{(j)}$ is independent of $(\nu_t^{(j)}, x_t^{(j)})$,

$$\mathrm{Cov}_{j,t}\left(\nu_t^{(j)} - \nabla f(x_t^{(j)}), \nu_{t+1}^{(j)} - \nu_t^{(j)}\right) = 0.$$

As a result,

$$\mathbb{E}_{j,t}\|\nu_{t+1}^{(j)} - \nabla f(x_{t+1}^{(j)})\|^2 = \|\nu_t^{(j)} - \nabla f(x_t^{(j)})\|^2 + \mathrm{Var}_{j,t}(\nu_{t+1}^{(j)} - \nu_t^{(j)}).$$

The proof is then completed by Lemma 4. $\qquad\square$

**Lemma 6.** *For any $j$,*

$$\mathbb{E}_j\|\nu_{N_j}^{(j)} - \nabla f(x_{N_j}^{(j)})\|^2 \leq \frac{m\eta^2 L^2}{b}\mathbb{E}_j\|\nu_{N_j}^{(j)}\|^2 + \frac{\sigma^2 I(B < n)}{B} + \frac{(d - k_1 - k_2)m}{k_2}R_j,$$

*where $\mathbb{E}_j$ is taken over all randomness in $j$-th outer loop (line 4-13 of Algorithm 1). 4.*

*Proof.* By definition,

$$\|\nu_{t+1}^{(j)}\| \leq \|\nu_t^{(j)}\| + \left\|\text{rtop}_{k_1,k_2}\left(M_t^{(j)}, \nabla f_{\mathcal{I}_t^{(j)}}(x_{t+1}^{(j)}) - \nabla f_{\mathcal{I}_t^{(j)}}(x_t^{(j)})\right)\right\|$$

$$\leq \|\nu_t^{(j)}\| + \left\|\nabla f_{\mathcal{I}_t^{(j)}}(x_{t+1}^{(j)}) - \nabla f_{\mathcal{I}_t^{(j)}}(x_t^{(j)})\right\|$$

$$\leq \|\nu_t^{(j)}\| + \frac{1}{b}\sum_{i\in\mathcal{I}_t^{(j)}}\left\|\nabla f_i(x_{t+1}^{(j)}) - \nabla f_i(x_t^{(j)})\right\|$$

$$\leq \|\nu_t^{(j)}\| + \sqrt{\frac{1}{b}\sum_{i\in\mathcal{I}_t^{(j)}}\left\|\nabla f_i(x_{t+1}^{(j)}) - \nabla f_i(x_t^{(j)})\right\|^2}$$

$$\leq \|\nu_t^{(j)}\| + \sqrt{\frac{2}{b}\left(\sum_{i\in\mathcal{I}_t^{(j)}}\left\|\nabla f_i(x_{t+1}^{(j)})\right\|^2 + \sum_{i\in\mathcal{I}_t^{(j)}}\left\|\nabla f_i(x_t^{(j)})\right\|^2\right)}$$

$$\leq \|\nu_t^{(j)}\| + \sqrt{\frac{2n}{b}\left(\frac{1}{n}\sum_{i=1}^n\left\|\nabla f_i(x_{t+1}^{(j)})\right\|^2 + \frac{1}{n}\sum_{i=1}^n\left\|\nabla f_i(x_t^{(j)})\right\|^2\right)}$$

$$\leq \|\nu_t^{(j)}\| + \sqrt{2n}\sigma$$

As a result,

$$\|\nu_t^{(j)}\| \leq \|\nu_0^{(j)}\| + t\sqrt{2n}\sigma, \tag{10}$$

Thus,

$$\|\nu_t^{(j)} - \nabla f(x_t^{(j)})\|^2 \leq 2\|\nu_t^{(j)}\|^2 + 2\|\nabla f(x_t^{(j)})\|^2 = \text{Poly}(t).$$

This implies that we can apply Lemma 2 on the sequence $D_t = \|\nu_t^{(j)} - \nabla f(x_t^{(j)})\|^2$.

Letting $j = N_j$ in Lemma 5 and taking expectation over all randomness in $\mathbb{E}_j$, we have

$$\mathbb{E}_j\|\nu_{N_j+1}^{(j)} - \nabla f(x_{N_j+1}^{(j)})\|^2$$

$$\leq \mathbb{E}_j\|\nu_{N_j}^{(j)} - \nabla f(x_{N_j}^{(j)})\|^2 + \frac{\eta^2 L^2}{b}\mathbb{E}_j\|\nu_{N_j}^{(j)}\|^2 + \frac{d - k_1 - k_2}{k_2}\mathbb{E}_j\left(g_{N_j}^{(j)} + \frac{G_{N_j}^{(j)}}{b}\right)$$

$$= \mathbb{E}_j\|\nu_{N_j}^{(j)} - \nabla f(x_{N_j}^{(j)})\|^2 + \frac{\eta^2 L^2}{b}\mathbb{E}_j\|\nu_{N_j}^{(j)}\|^2 + \frac{d - k_1 - k_2}{k_2}R_j. \tag{11}$$

By Lemma 2,

$$\mathbb{E}_j\|\nu_{N_j}^{(j)} - \nabla f(x_{N_j}^{(j)})\|^2 - \mathbb{E}_j\|\nu_{N_j+1}^{(j)} - \nabla f(x_{N_j+1}^{(j)})\|^2$$

$$= \frac{1}{m}\left(\|\nu_0^{(j)} - \nabla f(x_0^{(j)})\|^2 - \mathbb{E}_j\|\nu_{N_j}^{(j)} - \nabla f(x_{N_j}^{(j)})\|^2\right)$$

$$= \frac{1}{m}\left(\mathbb{E}_j\|\nu_0^{(j)} - \nabla f(x_{j-1})\|^2 - \mathbb{E}_j\|\nu_{N_j}^{(j)} - \nabla f(x_j)\|^2\right), \tag{12}$$

where the last line uses the definition that $x_{j-1} = x_0^{(j)}, x_j = x_{N_j}^{(j)}$. By Lemma 3,

$$\mathbb{E}_j\|\nu_0^{(j)} - \nabla f(x_{j-1})\|^2 \leq \frac{\sigma^2 I(B < n)}{B}. \tag{13}$$

The proof is completed by putting (11), (12) and (13) together. $\qquad\square$

**Lemma 7.** *For any $j, t$,*

$$f(x_{t+1}^{(j)}) \leq f(x_t^{(j)}) + \frac{\eta}{2}\|\nu_t^{(j)} - \nabla f(x_t^{(j)})\|^2 - \frac{\eta}{2}\|\nabla f(x_t^{(j)})\|^2 - \frac{\eta}{2}(1 - \eta L)\|\nu_t^{(j)}\|^2.$$

*Proof.* By (4),

$$\begin{aligned}
f(x_{t+1}^{(j)}) &\leq f(x_t^{(j)}) + \left\langle \nabla f(x_t^{(j)}), x_{t+1}^{(j)} - x_t^{(j)} \right\rangle + \frac{L}{2}\|x_t^{(j)} - x_{t+1}^{(j)}\|^2 \\
&= f(x_t^{(j)}) - \eta \left\langle \nabla f(x_t^{(j)}), \nu_t^{(j)} \right\rangle + \frac{\eta^2 L}{2}\|\nu_t^{(j)}\|^2 \\
&= f(x_t^{(j)}) + \frac{\eta}{2}\|\nu_t^{(j)} - \nabla f(x_t^{(j)})\|^2 - \frac{\eta}{2}\|\nabla f(x_t^{(j)})\|^2 - \frac{\eta}{2}\|\nu_t^{(j)}\|^2 + \frac{\eta^2 L}{2}\|\nu_t^{(j)}\|^2.
\end{aligned}$$

The proof is then completed. $\qquad\square$

**Lemma 8.** *For any $j$,*

$$\mathbb{E}_j\|\nabla f(x_j)\|^2 \leq \frac{2}{\eta m}\mathbb{E}_j(f(x_{j-1}) - f(x_j)) + \mathbb{E}_j\|\nu_{N_j}^{(j)} - \nabla f(x_j)\|^2 - (1 - \eta L)\mathbb{E}_j\|\nu_{N_j}^{(j)}\|^2,$$

*where $\mathbb{E}_j$ is taken over all randomness in $j$-th outer loop (line 4-13 of Algorithm 1).*

*Proof.* Since $\|\nabla f(x)\| \leq \sigma$ for any $x$,

$$|f(x_{t+1}^{(j)}) - f(x_t^{(j)})| \leq \sigma\|\nu_t^{(j)}\|.$$

This implies that

$$|f(x_t^{(j)})| \leq \sigma \sum_{k=0}^t \|\nu_t^{(j)}\| + |f(x_0^{(j)})|.$$

As shown in (10), $\|\nu_t^{(j)}\| = \mathrm{Poly}(t)$ and thus $|f(x_t^{(j)})| = \mathrm{Poly}(t)$. This implies that we can apply Lemma 2 on the sequence $D_t = f(x_t^{(j)})$.

Letting $j = N_j$ in Lemma 7 and taking expectation over all randomness in $\mathbb{E}_j$, we have

$$\mathbb{E}_j f(x_{N_j+1}^{(j)}) \leq \mathbb{E}_j f(x_{N_j}^{(j)}) + \frac{\eta}{2}\|\nu_{N_j}^{(j)} - \nabla f(x_{N_j}^{(j)})\|^2 - \frac{\eta}{2}\|\nabla f(x_{N_j}^{(j)})\|^2 - \frac{\eta}{2}(1 - \eta L)\|\nu_{N_j}^{(j)}\|^2.$$

By Lemma 2,

$$\mathbb{E}_j f(x_{N_j}^{(j)}) - \mathbb{E}_j f(x_{N_j+1}^{(j)}) = \frac{1}{m}\mathbb{E}_j(f(x_0^{(j)}) - f(x_{N_j}^{(j)})) = \frac{1}{m}\mathbb{E}_j(f(x_{j-1}) - f(x_j)).$$

The proof is then completed. $\qquad\square$

Combining Lemma 6 and Lemma 8, we arrive at the following key result on one inner loop.

**Theorem 3.** *For any $j$,*

$$\begin{aligned}
\mathbb{E}\|\nabla f(x_j)\|^2 \leq{}& \frac{2}{\eta m}\mathbb{E}_j(f(x_{j-1}) - f(x_j)) + \frac{\sigma^2 I(B < n)}{B} + \frac{(d - k_1 - k_2)m}{k_2}R_j \\
& - \left(1 - \eta L - \frac{m\eta^2 L^2}{b}\right)\mathbb{E}_j\|\nu_{N_j}^{(j)}\|^2.
\end{aligned}$$

### A.3 Complexity analysis

***Proof of Theorem 1.*** By definition (7) of $R_j$ and the smoothness assumption **A1**,

$$\mathbb{E}R_j \leq \frac{b+1}{b}L^2\mathbb{E}\|x_{N_j+1}^{(j)} - x_{N_j}^{(j)}\|^2 \leq 2\eta^2 L^2\mathbb{E}\|\nu_{N_j}^{(j)}\|^2.$$

By Theorem 3,

$$\mathbb{E}\|\nabla f(x_j)\|^2 \leq \frac{2}{\eta m}\mathbb{E}_j(f(x_{j-1}) - f(x_j)) + \frac{\sigma^2 I(B < n)}{B}$$
$$- \left(1 - \eta L - \frac{m\eta^2 L^2}{b} - \frac{2(d - k_1 - k_2)m\eta^2 L^2}{k_2}\right)\mathbb{E}_j\|\nu_{N_j}^{(j)}\|^2.$$

Since $\eta L = \sqrt{k_2/6dm}$,

$$\eta L + \frac{m\eta^2 L^2}{b} + \frac{2(d - k_1 - k_2)m\eta^2 L^2}{k_2} \leq \frac{1}{\sqrt{6}} + \frac{1}{6} + \frac{1}{3} \leq 1.$$

As a result,

$$\mathbb{E}\|\nabla f(x_j)\|^2 \leq \frac{2}{\eta m}\mathbb{E}_j(f(x_{j-1}) - f(x_j)) + \frac{\sigma^2 I(B < n)}{B}.$$

Since $x_{\text{out}} = x_{T'}$ where $T' \sim \text{Unif}([T])$, we have

$$\mathbb{E}\|\nabla f(x_{\text{out}})\|^2 \leq \frac{2}{\eta m T}\mathbb{E}(f(x_0) - f(x_{T+1})) + \frac{\sigma^2 I(B < n)}{B} \leq \frac{2\Delta_f}{\eta m T} + \frac{\sigma^2 I(B < n)}{B}.$$

The setting of $T$ and $B$ guarantees that

$$\frac{2\Delta_f}{\eta m T} \leq \frac{\epsilon^2}{2}, \quad \frac{\sigma^2 I(B < n)}{B} \leq \frac{\epsilon^2}{2}.$$

Therefore,

$$\mathbb{E}\|\nabla f(x_{\text{out}})\|^2 \leq \epsilon^2.$$

By Cauchy-Schwarz inequality,

$$\mathbb{E}\|\nabla f(x_{\text{out}})\| \leq \sqrt{\mathbb{E}\|\nabla f(x_{\text{out}})\|^2} \leq \epsilon.$$

In this case, the average computation cost is

$$\mathbb{E}C_{\text{comp}}(\epsilon) = T(\epsilon)\left(B + \frac{2(k_1 + k_2)}{d}bm\right) = 3BT(\epsilon)$$
$$= O\left(\frac{B\Delta_f}{\eta m\epsilon^2}\right) = O\left(\frac{\sqrt{Bb}L\Delta_f}{\epsilon^2}\sqrt{\frac{k_1 + k_2}{k_2}}\right).$$

The proof is then proved by the setting of $B$. □

***Proof of Theorem 2.*** Under the setting of $\eta$,

$$\eta L + \frac{m\eta^2 L^2}{b} \leq \frac{1}{\sqrt{3}} + \frac{1}{3} \leq 1.$$

By Theorem 3,

$$\mathbb{E}\|\nabla f(x_j)\|^2 \leq \frac{2}{\eta m}\mathbb{E}_j(f(x_{j-1}) - f(x_j)) + \frac{\sigma^2 I(B < n)}{B} + \frac{d - k_1 - k_2}{k_2}R_j.$$

By definition of $x_{\text{out}}$,

$$\mathbb{E}\|\nabla f(x_{\text{out}})\|^2 \leq \frac{2\Delta_f}{\eta m T} + \frac{\sigma^2 I(B < n)}{B} + \frac{(d - k_1 - k_2)m}{k_2}\mathbb{E}\bar{R}_T.$$

Under the settings of $T$ and $B$,

$$\frac{2\Delta_f}{\eta m T} \leq \frac{\epsilon^2}{3}, \quad \frac{\sigma^2 I(B < n)}{B} \leq \frac{\epsilon^2}{3}.$$

This proves the first result. The second result follows directly. For the computation cost, similar to the proof of Theorem 1, we have

$$\mathbb{E}C_{\text{comp}}(\epsilon) = O(BT) = O\left(\frac{L\Delta_f}{\epsilon^2}\frac{B}{\sqrt{m(b \wedge m)}}\right).$$

The proof is then completed by trivial algebra. □

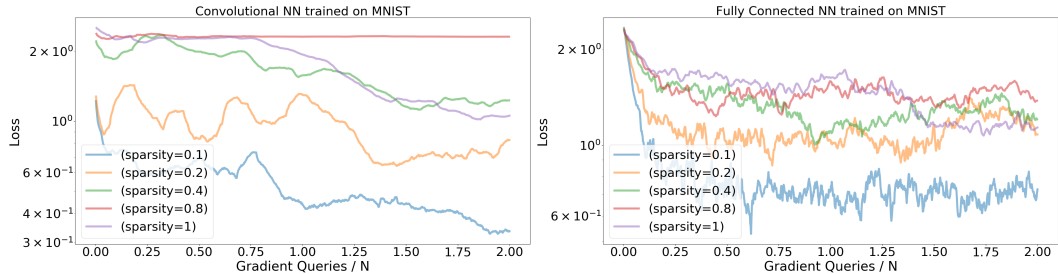

Figure 3: SpiderBoost with various values of sparsity, where (sparsity=$k/d$) corresponds to Spider-Boost with sparsity $k/d$. Both figures use MNIST. The x-axis measures gradient queries over $N$, where $N$ is the size of the respective datasets. Plots are in log-scale.

## B EXPERIMENTS

### B.1 DESCRIPTION OF SIMPLE CONVOLUTIONAL NEURAL NETWORK

The simple convolutional neural network used in the experiments consists of a convolutional layer with a kernel size of 5, followed by a max pool layer with kernel size 2, followed by another convolutional layer with kernel size 5, followed by a fully connected layer of input size $16 \times \text{side}^2 \times 120$ (side is the size of the second dimension of the input), followed by a fully connected layer of size $120 \times 84$, followed by a final fully connected layer of size $84\times$ the output dimension.

### B.2 NATURAL LANGUAGE PROCESSING

The natural language processing model consists of a word embedding of dimension 128 of 1000 tokens, which is jointly learned with the task. The LSTM has a hidden and cell state dimension of 1024.

---

**Algorithm 2:** SpiderBoost for Natural Language Processing.

**Input:** Learning rate $\eta$, inner loop size $m$, number of iterations $T$, large batch matrix $Z_2$ with $\ell_2$ batches of size $B$, small batch matrix $Z_1$ with $\ell_1$ batches of size $b$, initial iterate $x_0$, initial states $s_0$ and $S_0$.

1 **for** $t = 0, ..., T - 1$ **do**
2      $i = \text{mod}(t, \ell_1)$
3      $j = \text{mod}(t, \ell_2)$
4      **if** $i = 0$ **then**
5          $s_t = 0$
6      **if** $j = 0$ **then**
7          $S_t = 0$
8      **if** $\text{mod}(t, m) = 0$ **then**
9          $\nu_t, S_{t+1} := \nabla f_{Z_{2j}}(x_t, S_t)$
10          $s_{t+1} = s_t$
11      **else**
12          $g_p := \nabla f_{Z_{1i}}(x_{t-1}, s_{t-1})$
13          $g_c, s_{t+1} := \nabla f_{Z_{1i}}(x_t, s_t)$
14          $\nu_t := \nu_{t-1} + (g_c - g_b)$
15          $S_{t+1} = S_t$
16      $x_{t+1} := x_t - \eta \nu_t$

**Output:** $x_T$

---

Before describing Algorithm 2, let us derive the full batch gradient of a generative language model. We encode the vocabulary of our dataset of length $N$ so that $D \in \mathbb{N}^N$ is a sequence of integers corresponding to one-hot encodings of each token. We model the transition $p(D_{i+1}|D_i, s_i)$ using an RNN model $M$ as $M(D_i, s_i) = D_{i+1}, s_{i+1}$, where $s_i$ is the sequential model state at step $i$. The

model $M$ can be thought of as a classifier with cross entropy loss $L$ and additional dependence on $s_i$. The batch gradient objective can therefore be formulated by considering the full sequence of predictions from $i = 0$ to $i = N - 1$, generating for each step $i$ the output $\hat{D}_{i+1}, s_{i+1}$. Each token is one-hot encoded as an integer (from 0 to the size of the vocabulary), so the empirical risk is given by

$$J(D; x) = \frac{1}{N} \sum_{i=0}^{N-1} L(\hat{D}_i, D_i).$$

Thus, the full batch gradient is simply the gradient of $J$ with respect to $x$.

In Algorithm 2, $D$ is split into $b$ contiguous sequences of length $\ell_1 = N/b$ and stored in a matrix $Z_1 \in \mathbb{N}^{b \times \ell_1}$. Taking a pass over $Z_1$ requires maintaining a state $s_i \in \mathbb{N}^b$ for each entry in a batch, which is reset before every pass over $Z_1$. To deal with maintaining state for batches at different time scales, we define a different matrix $Z_2 \in \mathbb{N}^{b \times \ell_2}$ which maintains a different set of states $S_i \in \mathbb{N}^B$ for each entry of batch size $B$. We denote by $g, s_{t+1} = \nabla f_{Z_{1j}}(x, s_t)$ the gradient of our model with respect to $x$, where $\nabla f_{Z_{1j}}$ denotes the gradient function corresponding to the $j$th batch of matrix $Z_1$. The function $f_{Z_{1j}}$ simply computes the loss of the $j$th batch of matrix $Z_1$.

