# OpenReview forum: "Variance Reduction With Sparse Gradients"
_ICLR.cc/2020/Conference — Accept (Poster)_

### Official Review · AnonReviewer2 · 2019-10-14
**Official Blind Review #2**

**Rating:** 3

**Review:**

Summary:
The author(s) provide a method which combines some property of SCGS method and SpiderBoost. Theoretical results are provided and achieve the state-of-the-art complexity, which match the one of SpiderBoost. Numerical experiments show some advantage compared to SpiderBoost on some deep neural network architecture for some standard datasets MNIST, SVHN, and CIFAR-10.

Comments:

1) It is true that variance reduction methods achieve the state-of-the-art complexity theory for finding first order stationary point of general nonconvex optimization problems. However, it is well-known that variance reduction methods are not very efficient for training deep neural networks. All of the experiments in this paper are focusing on deep learning problems. If the author(s) would like to show good performance, I would suggest to compare the algorithms with the state-of-the-art algorithms in Deep Learning such as Adam, SGD-Momentum. Showing some improvement over SpiderBoost for deep learning problems would have low impact.

2) I would suggest the author(s) to switch directions to focus on general nonconvex problems, that is, to find some different examples on general non-convex optimization problems rather than for deep learning problems. In other words, to find examples which show that your algorithm has more advantage than SGD-type algorithms, SVRG-type.

3) I would also suggest the author(s) to plot all figures in log-scale in order to see in more detail performance.

4) According to my knowledge, SpiderBoost is an alternative way of re-writing the SARAH algorithm [1, 2] with some small modification, that is a variant of SARAH. Therefore, the SARAH algorithm should be highly related to this paper and need to be discussed and mentioned more clearly.

[1] Nguyen et al 2017a, “SARAH: A Novel Method for Machine Learning Problems Using Stochastic Recursive Gradient”.
[2] Nguyen et al 2017b, “Stochastic Recursive Gradient Algorithm for Nonconvex Optimization”.


**Experience Assessment:**

I have published in this field for several years.

**Review Assessment: Checking Correctness Of Derivations And Theory:**

I carefully checked the derivations and theory.

**Review Assessment: Checking Correctness Of Experiments:**

I carefully checked the experiments.

**Review Assessment: Thoroughness In Paper Reading:**

I read the paper thoroughly.

---

> ### Author Response · Authors · 2019-11-12
> **Reply to Review #2**
>
> Thank you for your feedback. Indeed, some implementations of SVRG-style variance reduction methods are shown to be ineffective for training deep neural networks (Defazio et al. 2018). We have reproduced these results and acknowledge these findings in our paper with the appropriate citations. The covariance between the gradient snapshot computed every $m$ iterations and the gradient computed in the inner loop decreases rapidly for very large deep learning models. Our algorithm reduces the cost of computing variance reduction terms, which reduces the number of inner loop iterations required to see the benefits of the variance reducing computations. Furthermore, by using sparsity, our algorithm naturally reduces the overall potential variance of these methods. This potential variance occurs when the aforementioned covariance approaches zero. Our algorithm is only beneficial when there is sparsity structure in the gradients of the objective, and we experimentally show that such sparsity structure exists in the gradients of some deep neural networks.
>
> We view this work as taking a step toward making SVRG/SCSG-style variance reduction methods practical. These methods provide a unique way to estimate gradient sparsity by taking advantage of the full/large-batch gradient computed at the beginning of each outer-loop. In other words, the correction term in SVRG/SCSG-style algorithms can be used for two purposes at the same time without extra overhead: Variance reduction and estimation of gradient sparsity. By contrast, this cannot be done with SGD-style algorithms without introducing significant computational overhead.
>
> While this work can be viewed as addressing some of the observations made by Defazio et al., we plan to directly address them in future work. Lei et al. (2017) comment that the mechanism used by SVRG/SCSG-style variance reduction methods to accelerate gradient-based methods is qualitatively different than momentum (in SGD-momentum/Adam) and adaptive stepsizes (in AdaGrad/Adam). We believe that, if properly combined with existing state-of-the-art techniques, SVRG/SCSG-style variance reduction methods have the potential to further reduce stochastic variance.
>
> - Experiments: To concretely address your feedback about our experiments, we will add at least one additional non-convex problem which exhibits gradient sparsity.
>
> - Figures in Log-scale: Thank you for the suggestion. We updated our figures to log-scale.
>
> - Connection With SARAH: Thank you for pointing out our missing references. We added them into our references and discussed them in Section 2.

---

> > ### Comment · AnonReviewer2 · 2019-11-14
> > **Response**
> >
> > I do agree that this paper has some theoretical contributions related to sparsity for general nonconvex problems.
> >
> > - Please add experiments on general non-convex example to show the advantages of your proposed methods with others (not just only with SpiderBoost -- I believe that in term of numerical experiments, SpiderBoost is not the best one).
> > - Please revise your related work properly.
> > - Please highlight your contributions in the introduction part.
> >
> > I will update my score accordingly.

---

### Official Review · AnonReviewer1 · 2019-10-23
**Official Blind Review #1**

**Rating:** 6

**Review:**

This paper aims at improving the computational cost of variance reduction methods while preserving their benefits regarding the fast provable convergence. The existing variance reduction based methods suffer from higher per-iteration gradient query complexity as compared to the vanilla mini-batch SGD, which limits their utility in many practical settings. This paper notices that, for many models, as the training progresses the gradient vectors start exhibiting structure in the sense that only a small number of coordinates have large magnitude. Based on this observation, the paper proposes a modified variance reduction method (by modifying the SpiderBoost method), where a 'memory vector' keeps track of the coordinates of the gradient vectors with large variance. Let $d$ be the size of the model parameter. During each iteration, one computes the gradient for $k_1$ coordinates with the highest variance (according to the memory vector) and an additional $k_2$ random coordinates.

The paper shows that in the worst case, the proposed method has the same gradient query complexity as the SpiderBoost variance reduction method. Assuming that the proposed method can track the sparsity of the gradient vector, the proposed method achieves a gradient query complexity which is $O(\sqrt{(k_1 + k_2)/d})$ times that of the SpiderBoost method. The paper demonstrates the gradient query complexity improvement over the SpiderBoost method on MNIST and CIFAR-10 data set.

The paper presents novel results by utilizing the ideas from the field of communication-efficient distributed optimization. As far as the reviewer can tell, the results in the paper are correct. That said, there is quite a bit of room for improvement in terms of the writing of the paper.

The paper appears to have way too many typos. For example,

In Section 2.1:
- Why is $k$ introduced?
- $S$ denotes a random subset with size $k$ ---> $k_2$?
- drawn from the set ${\ell : |y_{\ell}| < |y_{(k)}|}$ ----> ${\ell : |x_{\ell}| < |x_{(k_2)}|}$?
- $rtop(x, y) = (0, 12, 0, 0, 1)$ --> $rtop(x, y) = (0, 16, 0, 0, 1)$
In Lemma 1:
 - while defining $top_{-k_1}(x, y)$, ".... if |x_{\ell}| >= |x_{(k_1)}|" ----> ".... if |x_{\ell}| <= |x_{(k_1)}|"?
In Section 2.2:
 - What are $g_0, g_1,..., g_{L-1}$? Shouldn't these be $\phi_0, \phi_1,..., \phi_{L-1}$?
In A1:
 - right after (5), what is $\tilde{x}_0$ in the definition of $\Delta_f$?

The authors may also consider making the empirical evaluation more comprehensive by considering tasks from the NLP domain, e.g., language modeling. This would further help asses the utility of the proposed method.

**Experience Assessment:**

I have read many papers in this area.

**Review Assessment: Checking Correctness Of Derivations And Theory:**

I assessed the sensibility of the derivations and theory.

**Review Assessment: Checking Correctness Of Experiments:**

I assessed the sensibility of the experiments.

**Review Assessment: Thoroughness In Paper Reading:**

I read the paper at least twice and used my best judgement in assessing the paper.

---

> ### Author Response · Authors · 2019-11-12
> **Reply to Review #1**
>
> Thank you for your thoughtful feedback. Please find below our responses to your comments.
>
> - Typos in Section 2.1: We apologized for the typos. In the new draft, we updated definitions and examples for corrections and enhanced clarity.
>
> - Section 2.2 Definition of Activation Functions: This is now simplified to use only $\phi$ instead of introducing $g$.
>
> - A1: what is $\overline{x}_0$ in $\Delta f$: $\tilde{x}_{0}$ should be $x_{0}$. We correct it in the updated draft.
>
> - We are currently looking into adding additional experiments to address your suggestion to broaden the variety of experiments.

---

### Official Review · AnonReviewer3 · 2019-10-23
**Official Blind Review #3**

**Rating:** 8

**Review:**

Summary: This paper introduces a sparse variant to SpiderBoost which reduces the complexity cost of updating gradient estimates by way of sparse updates. The authors prove that this variant incurs a negligible increase in worst case complexities as soon as certain assumptions are satisfied, and that when their algorithm captures sparsity correctly, they improve upon SpiderBoost's complexity.

This paper is clearly, and the experiments support the theoretical contributions.

In Figure 1, you report results as a function of gradient queries/N. Given Theorem 2, I assume that the graphs would look similar as a function of wall-clock time; can you confirm this?

Recommendation: Accept.

Minor comments and questions for the author:
- I am slightly confused by the introduction of the rtop operator. Specifically,
  1) What is the relation between k1, k2, and k?
  2) You write that S is a random subset of size k. Should this be k2?
  3) In your first example, should we have rtop(x,y) = (0, 16, 0, 0, 1), since for \ell = 2, y_\ell = 4, d-k1 = 4, k2=1? Am I missing something?
  More generally, my understanding is that the rtop(x,y) operator randomly sparsifies y based on x, which essentially provides indication of where sparsity would be least harmful; when not sparsifying, rtop applies a rescaling that guarantees unbiased estimates. If this is correct, I would recommend making that intuition more clear early on in the paper, in order to improve upon the clarity of the paper.

- You state that rtop is linear in y; since rtop depends on the random variable S, is the claim that E[rtop(x, y+y')] = E[rtop(x, y)]+E[rtop(x,y')] (which follows from unbiasedness)?

- For your experiments, could you discuss how your choice of hyperparameters relates to the constraints in Theorem 1 and 2?

- I believe Table 1 would be more impactful if it also included the initial entropy ratios at the beginning of training, rather than reporting those values below.

- Other variance reduction techniques for minibatching focus on choosing the minibatches themselves with non-uniform sampling. Under such a sampling mechanism, do you foresee any complications to using SpiderBoost with Sparse Gradients, eg., decrease in overall sparsity?

**Experience Assessment:**

I have read many papers in this area.

**Review Assessment: Checking Correctness Of Derivations And Theory:**

I assessed the sensibility of the derivations and theory.

**Review Assessment: Checking Correctness Of Experiments:**

I carefully checked the experiments.

**Review Assessment: Thoroughness In Paper Reading:**

I read the paper at least twice and used my best judgement in assessing the paper.

---

> ### Author Response · Authors · 2019-11-12
> **Reply to Reviewer #3**
>
> Thank you for your thoughtful feedback. Please find below our responses to your comments.
>
> - queries/N and wallclock time: We are currently looking into adding additional experiments to address this question.
>
> - Confusion about rtop operator: We apologize for the typos here and thank you for providing a more intuitive explanation. We updated the paper to address confusion about the definitions of $k$, $k_1$, and $k_2$. We also updated definitions and examples for corrections and enhanced clarity. We added a more intuitive explanation of what the rtop operator does before formally introducing the operator.
>
> - Linearity of rtop Operator: By linearity we mean that rtop(x, y+y') = rtop(x, y) + rtop(x, y') for a fixed S and hence E[rtop(x, y+y')] = E[rtop(x, y)]+E[rtop(x,y')]. We added a sentence clarifying this.
>
> - Theoretical Motivation of Hyperparameters: Based on Theorems 1 and 2, Our experiments are carried out by specifying the fraction $k/d$ of gradient coordinates we want used for variance reduction. We then set set $k_1 = k // 2$, where $//$ is integer division, and $k_2 = k - k_1$. In our experiments, we set $k/d = 0.1$. We set $B = c b$ for a small constant $c$ (in our experiments we use $c=10$). We then set $m = B/b$.
>
> - Impact of non-uniform minibatch sampling: Non-uniform minibatch sampling has no negative effect on sparsity because it only changes the probability that each entry is sampled and not the sparsity pattern.

---

### Author Response · Authors · 2019-11-12
**Thank You**

We thank all reviewers for their thoughtful comments. We also thank them for pointing out typos and potential issues on clarity. We will revise our paper twice before discussions end. The first revision, which we have already submitted, addresses all the issues noted in our response to each reviewer. Our second and final revision adds experiments based on the suggestions of each reviewer.

---

### Author Response · Authors · 2019-11-15
**Summary of Updates to Second Revision**

Our most recent submission includes the following notable changes:

1. We have added a natural language processing experiment in response to Reviewer #1. This is a generative LSTM language model which involved additional algorithmic design decisions to deal with intermediate states at different time scales. The model is trained on the Penn Treebank corpus. This is compared to an SGD baseline with constant learning rate.

2. We have added a sparse matrix factorization experiment in response to Reviewer #2. This is a Bayesian Personalized Ranking model train on the 100k MovieLens dataset. We implement a learning rate schedule for our algorithm, which we consider a fair analogue to an exponential decay learning rate schedule for SGD. The SGD baseline for this experiment uses an exponential decay learning rate schedule.

3. We have updated our related work, and added a subsection which more clearly defines our contributions.

We were planning to include an experiment to demonstrate an implementation of sparse back propagation for wall-clock time comparison in response to Reviewer #3, but we ran out of time.

We'd like to thank the reviewers, area chairs, and anyone else involved in the decision making process for their time and consideration.

---

### Decision · Program_Chairs · 2019-12-19

**Decision:**

Accept (Poster)

**Comment:**

Congratulations on getting your paper accepted to ICLR. Please make sure to incorporate the reviewers' suggestions for the final version.